# Flip-Flap: A Simple Dual-View Imaging Method for 3D Reconstruction of Thick Plant Samples

**DOI:** 10.3390/plants11040506

**Published:** 2022-02-13

**Authors:** Leo Serra, Sovanna Tan, Sarah Robinson, Jane A. Langdale

**Affiliations:** 1The Sainsbury Laboratory, University of Cambridge, Cambridge CB2 1LR, UK; sarah.robinson@slcu.cam.ac.uk; 2Department of Plant Sciences, University of Oxford, South Parks Rd., Oxford OX1 3RB, UK; jane.langdale@plants.ox.ac.uk

**Keywords:** plant 3D imaging, multiview imaging, rice, barley, Marchantia

## Abstract

Plant development is a complex process that relies on molecular and cellular events being co-ordinated in space and time. Microscopy is one of the most powerful tools available to investigate this spatiotemporal complexity. One step towards a better understanding of complexity in plants would be the acquisition of 3D images of entire organs. However, 3D imaging of intact plant samples is not always simple and often requires expensive and/or non-trivial approaches. In particular, the inner tissues of thick samples are challenging to image. Here, we present the Flip-Flap method, a simple imaging protocol to produce 3D images of cleared plant samples at the organ scale. This method allows full 3D reconstruction of plant organs suitable for 3D segmentation and further related analysis and can be easily handled by relatively inexperienced microscopists.

## 1. Introduction

Plants are complex three-dimensional (3D) structures. Visualizing the spatial distribution of cellular features (geometry, fate, localization) and molecular components with high resolution in 3D is essential to the study of many aspects of plant biology. However, many plant tissues are difficult to image. For example, some species can be highly auto-fluorescent due to cellular compounds or organelles such as chloroplasts [1]. The composition and organization of the cell wall, which affects fluorescence acquisition, can also differ between cell types and developmental stages, as well as between plant species. In addition, the plant cell wall induces much light scattering, which is detrimental for light-based microscopy [1,2]. Rice leaves in particular have a dense waxy cuticle that scatters light, as well as highly auto-fluorescent compounds that make imaging challenging even when specifically modified fluorescent reporter proteins are used [3]. The variability of these issues between species, and also within species between plant organs and during organ growth, makes some imaging experiments very challenging.

The outermost epidermal layer or overall organ shape can readily be observed even for big samples of plant tissue, using either fluorescent-based or scanning electron microscopes (SEM), but histological sections remain essential to gain information on the expression patterns and cellular organization of the internal tissues. Notably, tissue sectioning does not preserve the 3D structure of samples, limiting the reliability of any data obtained this way. To circumvent this drawback, protocols combining fixation, clearing, and staining have been developed to enable 3D imaging of entire plant organs [4,5,6,7]. These protocols have been successfully used to produce 3D cell atlases of organs such as Arabidopsis roots, meristems, and ovule primordia [8,9], but they do not allow deep imaging of samples over 100 µm. Observations of bigger samples are usually performed using X-ray computed tomography, a method that enables full 3D imaging of samples such as the root system of a maize plant (reviewed in [10]). However, despite recent improvements, the quality of the signal in this method does not allow accurate 3D segmentation, and it is not suitable for fluorescently labelled samples [11]. A promising approach would be to couple clearing protocols with multiview imaging methods such as light sheet microscopy, as has been used in animal systems such as the mouse brain [12]. However, light sheet microscopy requires a specialised setup, which is not widely available, and the sample mounting and imaging are not trivial. In contrast, laser scanning confocal microscopes (LSCMs) are widely available, require little training for users, and have already been successfully used to image Arabidopsis meristems [13]. Here, we present the Flip-Flap method, a simple imaging protocol suitable for any standard LSCM that enables dual-view imaging of cleared and stained thick plant samples (up to 300 µm in thickness), allowing full 3D reconstruction.

## 2. Results

To develop a new pipeline for imaging plant tissues, we first considered the limitations of existing methods. Standard sample mounting procedures for LSCM imaging involve placing an object between a slide and a coverslip. When cleared and stained plant samples are imaged in this way, a good signal is obtained in the outer 100–150 µm of the sample, but deeper in, the signal quality is reduced. Because 150 µm is halfway through a 300 µm thick sample, we reasoned that imaging the same sample from both sides would enable full 3D reconstruction. To this end, we first mounted samples in a “coverslips sandwich” using Tough tag© as spacers and a glycerol-based mounting medium. We then designed and 3D printed a holder for the sandwich (sandwich holder) (Figure 1A). This mounting method enabled two stacks of the same sample to be acquired with a 180-degree rotation between the two. The two stacks (Flip and Flap) were then processed through a simple pipeline to obtain a 3D reconstruction. First, one of the stacks was flipped around the x/y axis and around the z axis, so that the two stacks were oriented the same way. The two were then stitched together (Figure 1B; see also the Section 4 for a detailed procedure in ImageJ). Since the pairwise stitching plugin [14] used to stitch the two stacks only performs translations in the x, y, and z axes, any rotation in one or more of the axis will lead to artifacts in the stitching. The best way to prevent rotation in the z axis is to adjust the number of spacers to the thickness of the sample (1 Tough-Tag^TM^ is approximately 50 µm in thickness). Small x/y rotations might occur after flipping the sample holder upside down. These can be manually corrected using the rotation function in ImageJ (Appendix A). This Flip-Flap pipeline enabled us to obtain a full 3D image of a vegetative shoot apex of barley (shoot apical meristem plus Plastochron 1, 2, and 3 leaf primordia) (Figure 1C,D).

The thickness and properties of plant tissues can vary significantly between plant organs and species. Therefore, to further test the power of the Flip-Flap method, we attempted to image a range of organs from different plant species, including monocots, which are known to be challenging for fluorescence imaging. The Flip-Flap method enabled intact monocot leaf primordia to be imaged with cellular resolution (Figure 1C,D; Figure 2), allowing subsequent 3D reconstruction. Similarly, the method provided images through the entire thickness of a mature barley leaf and a gemma of the liverwort *Marchantia polymorpha* (Figure 2)—two samples that to our knowledge have not previously been imaged in 3D through the whole sample. The most challenging sample was the rice root. Although we managed to reconstruct the root in 3D, the diminished fluorescence signal from the vascular stele would likely compromise any downstream analysis. This limitation could probably be overcome by optimizing the time of incubation in fluorescent dye or by using plasma membrane fluorescent reporter lines compatible with clearing. With appropriate optimization, the method is thus applicable in most contexts where a 3D image of plant tissues less than 300 µm thick is required.

Finally, to assess whether the Flip-Flap method allows 3D reconstruction suitable for segmentation, the MorphoGraphX software [15] was used to analyse images of rice samples (Figure 3; Appendix A). After loading the stitched stack into the software, a Gaussian blur was applied to reduce noise before performing a watershed autoseeded 3D segmentation. As shown in Figure 3, this approach segmented the cells in 3D through the whole sample thickness, enabling downstream analysis. The segmented image provided data such as cell volume, cell axis length, and fluorescent protein localization at the whole organ scale.

Taken together, these results show that the Flip-Flap method is a versatile tool that allows 3D reconstruction of whole thick samples from different plant species and organs, facilitating subsequent analysis of biological processes at the cellular, tissue, and even organ scale.

## 3. Discussion

Advances in biological sample preparation and imaging technologies have greatly improved our knowledge of biological processes and organ development over the last decade, and in particular, tools to acquire and analyse 3D images of biological samples have allowed more in-depth analysis of animal and plant organs [15,16,17,18]. However, most approaches are still limited to small samples and/or tissues with low light scattering, and many require expensive microscopy equipment. In particular, plant tissues remain difficult to image, in part due to the variable composition of the cell wall between different plant species and between different plant tissues within any given species. The Flip-Flap method presented here allows 3D imaging of large plant organs through dual-view imaging on a “standard” confocal microscope followed by reconstruction of a complete z stack with image processing software. Image processing is a crucial step for successful 3D reconstruction. Indeed, misalignments between the Flip and Flap stacks can prevent stitching, or at least greatly decrease the quality of stitching, leading to artefacts that prevent further 3D analysis. In these circumstances, manual correction may be necessary. Alternatively, corrections can be achieved using registration plugins that are freely available for ImageJ (the list of which can be found on the ImageJ website—https://imagej.net/imaging/registration—(accessed on 7 February 2022) with all their characteristics). The stitched stacks are suitable for 3D segmentation and related analyses. The use of different clearing methods such as Clearsee, Clearsee-alpha, or PEA-clarity could be optimized for different tissues to improve image quality [4,5,19] (reviewed in [20]), and the use of multi-photon LSCM modules could enhance images of deep tissues. However, this method is still limited to fixed and cleared tissue and is therefore not suitable for live tissue imaging. In addition, as with other multiview imaging approaches, Flip-Flap datasets require much digital storage space and a high level of computational power for analysis. Undoubtedly, the computational limitations can be overcome, as for example with the visualization and analysis of the Drosophila brain, where a high-performance computing pipeline was developed and combined with a BigDataViewer-based viewer plugin to assemble thousands of 3D image files [17,21]. In the meantime, the Flip-Flap method can be applied to a wide range of plant samples.

The processing and analysis of 3D datasets with convolutional neural networks such as PlantSeg to predict cell boundaries [22], combined with 3D segmentation and cell type predictions in MorphoGraphX [18], provide a very powerful way to generate 3D cell atlases of plant organs. In the absence of protocols to perform time-lapse imaging of plant organs with good resolution of inner tissues, the Flip-Flap method provides the best method to collect such datasets. This advance paves the way to answer previously intractable questions about how the development of different tissues and cell types within an organ are inter-related, both spatially and temporally. For example, several models and studies suggest that the patterning of inner leaf tissues is interconnected with the patterning of the leaf epidermis, with the position of veins (reviewed in [23]) and air spaces (reviewed in [24]) thought to influence stomatal positioning. Flip-Flap will allow these questions to be addressed in intact organ primordia, even in monocot species where previous studies have been limited to tissue sectioning or surface imaging. Another potential application of the Flip-Flap method is to generate 3D templates for models. Indeed, real 3D cell geometries extracted from images have been recently used to predict the orientation of cell divisions, microtubules, and physical stressors [25,26,27]. In conclusion, the use of the Flip-Flap method combined with optimized fluorescent molecular markers [3] will enable comprehensive 3D atlases of cell properties and gene expression profiles to be built in developing organs of even the most challenging plant species.

## 4. Materials and Methods


**Plant material and growth conditions:**


Dehulled rice seeds (*Oryza sativa spp. japonica* cultivar Kitaake) were surface sterilized with 70% ethanol for 5 min followed by a 15 min wash in 25% sodium hypochlorite (Fisher Scientific, Hampton, NH, USA), 0.1% Tween (Sigma, St. Louis, MO, USA), and then, 5 washes in sterile water. Seeds were sown on half-MS (pH 5.8) plates supplemented with 1.5% sucrose and incubated in a growth cabinet (Panasonic MLR-352-PE) at 30 °C/25 °C, 16 h/8 h day, 150–200 µmol photons m^2^ s^−1^/night for 4 d to 6 d before being dissected.

Barley “Golden promise” cultivar seeds were surface sterilized in 95% ethanol, then dried and sown on plates containing water and 1% agarose, stratified at 4 °C in the dark for 4 d, then grown for 6 d in standard long photoperiod conditions (16 h light/8 h dark at 20 °C).

Gemmae of *Marchantia polymorpha* ecotype TAK-1 were collected from a gemma cup on a 4-week-old thallus that was grown on 1/2 Gamborg’s medium (pH 5.7, 1% sucrose without B5 vitamins) under continuous light at 22 °C.


**Fixation, clearing, and staining:**


After dissection, rice plantlets were placed on plates containing half-MS supplemented with 1.5% (*w*/*v*) sucrose and 0.1% (*v*/*v*) Plant Preservative Mixture (PPM™; Plant Cell Technology) for 16 h at 30 °C in the dark. After this recovery period, plantlets were fixed with 90% ice-cold acetone for 10 min followed by a 2 h incubation in 4% *v*/*v* formaldehyde, 5% *v*/*v* acetic acid, and 50% *v*/*v* ethanol (FAA) at room temperature. Samples were then washed 3 times in phosphate-buffered saline (PBS) pH 7 before clearing. Barley and Marchantia samples were not fixed prior to clearing. Samples were all cleared in Clearsee (10% xylitol (*w*/*v*), 15% sodium deoxycholate (*w*/*v*), and 25% urea (*w*/*v*) [4] with the exception of Marchantia, which was cleared in Clearsee-alpha CS5 (Clearsee plus 50 mM sodium sulphite) [5]. Clearing was carried out for 5 days to 2 weeks, depending on the tissue type. The cell walls of rice leaf primordia were stained by adding 0.1% (*w*/*v*) Direct Red 23 into the clearing solution. To stain the cell walls in rice roots, barley, and Marchantia samples, calcofluor was added to the clearing solution (0.1% for rice or 0.01% for barley and Marchantia—*v*/*v* from a 10% stock solution in 70% EtOH). Samples were rinsed and stored in Clearsee solution at 4 °C until imaging.


**Design and printing of the sandwich holder:**


The sandwich holder was designed using the free Tinkercad online software ((https://www.tinkercad.com) accessed on 7 February 2022) and was printed on a Ultimaker S5 3D printer using Cura Ultimaker software to define slicing prior to printing.

The .stl file of the holder has been deposited in the Mendeley database at doi: 10.17632/bp6f3bcj5j.1.


**Imaging:**


Samples were mounted as described in Figure 1 in a medium of 75% glycerol and 25% PBS (pH7). Samples were imaged on a Leica SP8 or a Zeiss LSM880 using 20× oil immersion (Leica HC PL APO 20×/0.75) and 25× glycerol immersion (LD LCI Plan-Apochromat 25×/0.8) objectives, respectively. Calcofluor was excited at 405 nm, and emission was collected between 415 nm and 480 nm. Direct Red was excited at 561 nm, and emission was collected between 575 nm and 615 nm. All images were acquired with a maximal pixel size of 500 nm in x/y, as well as a maximal z step of 500 nm.


**Image processing and visualization:**


Deconvolution was performed on rice image stacks using the Regularized Inverse Filter defaults method using ZENblue 2.3 software. Prior to stitching, the Flap stack was flipped vertically/horizontally and the z order reverted (image > transform > flip horizontally; image > transform > flip z) using ImageJ ((https://imagej.nih.gov/ij/) accessed on 7 February 2022) [28]. A region of interest was drawn on a feature shared by the 2 stacks using the rectangle selection tools. Finally, stitching of the 2 image stacks was carried out using the ImageJ pairwise stitching plugin [14] with default parameters. 3D visualization of images was carried out using MorphoGraphX software [15] ((https://morphographx.org/) accessed on 7 February 2022).

The MorphoGraphX 3D segmentation pipeline was as follows: a Gaussian blur was applied to the reconstructed 3D stacks (process > stack > filter > Gaussian Blur Stack 0.3 um in x, y, and z), then the blurred stack was segmented using the ITK watershed autoseeded process with default parameters (process > stack > ITK > ITK watershed autoseeded), and finally, the outer label was deleted (process > stack > segmentation > erase at border), and the segmentation was manually corrected.

## Figures and Tables

**Figure 1 plants-11-00506-f001:**
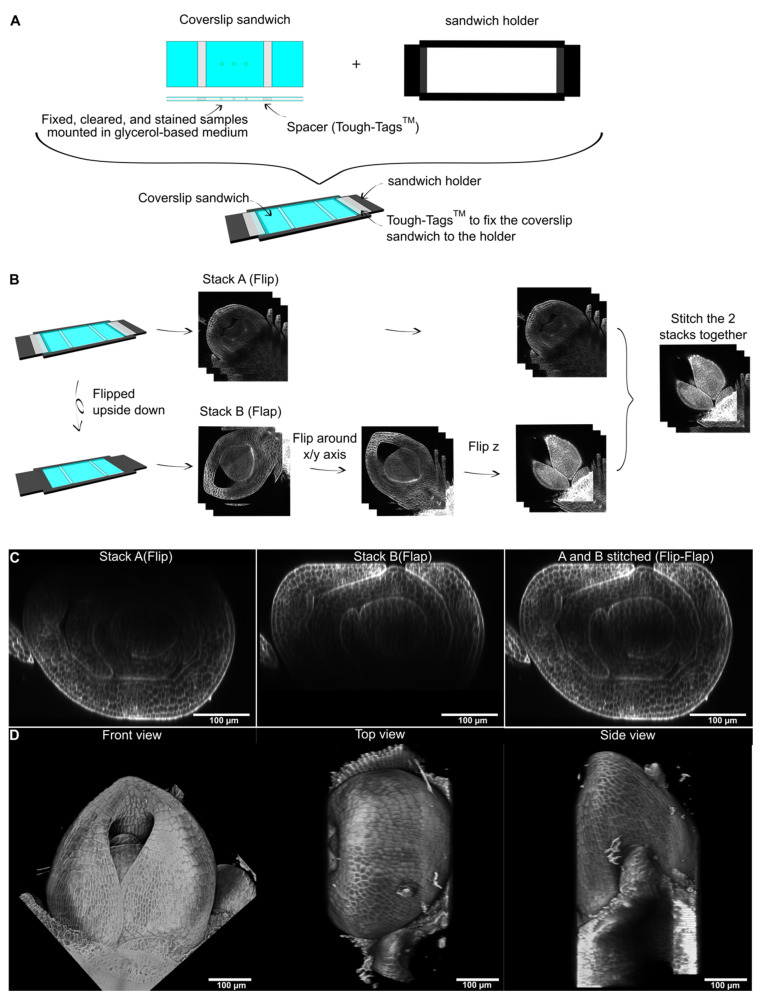
**Description of the Flip-Flap method for 3D reconstruction of plant samples.** (**A**) Fixed, cleared, and stained samples are mounted between two coverslips in a glycerol-based medium with spacers to prevent the sample from being squashed (coverslip sandwich). This mount is then attached to a 3D-printed holder (sandwich holder). (**B**) The mount is imaged from both sides using a laser scanning confocal microscope (LSCM) to produce two z series (Stack A/Flip and Stack B/Flap). Stack B is then rotated around the x or y axis, and the z order is inverted, before sticking Stack A to Stack B to produce the full 3D image. (**C**) Transverse optical section through Flip, Flap, and a stitched image of a barley shoot apex stained with calcofluor. (**D**) Front, top, and side view of the 3D reconstructed apex. Scale bars: 100 µm.

**Figure 2 plants-11-00506-f002:**
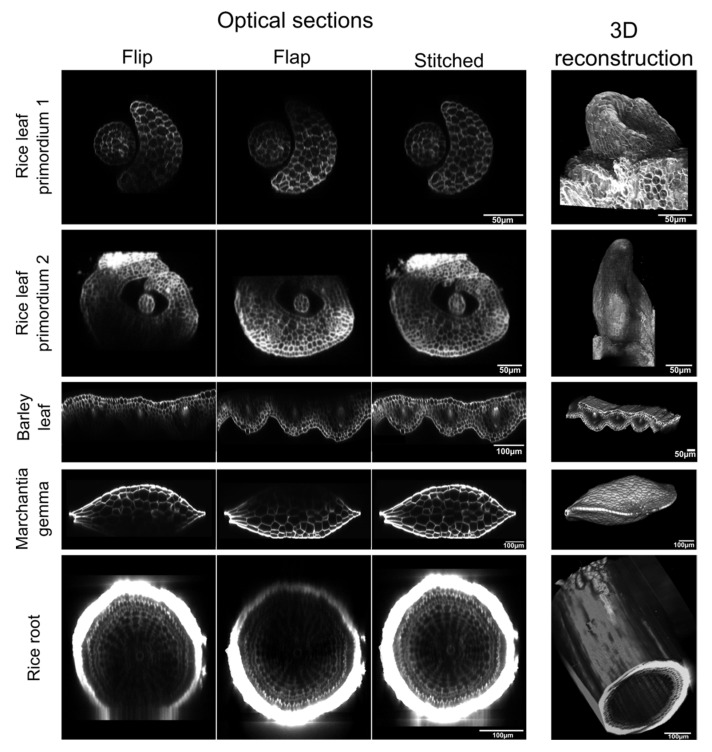
**Optical sectioning of single or stitched stacks in different plant tissues.** Visualization of optical cross-sections in Flip, Flap, and stitched stacks, along with the resulting 3D reconstructions generated using MorphoGraphX software. The samples were stained with Direct Red 23 (rice leaf primordia) or calcofluor (barley leaf, Marchantia gemma, and rice roots) according to established protocols. Scale bar: 50 µm (rice and barley leaves); 100 µm (Marchantia gemma and rice roots).

**Figure 3 plants-11-00506-f003:**
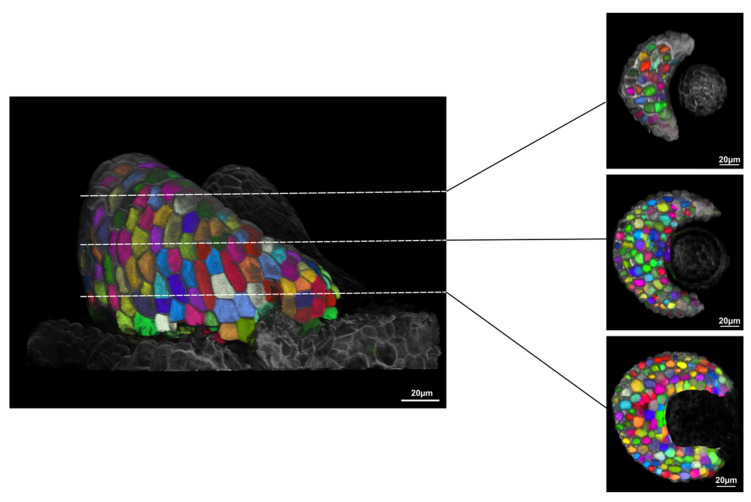
**Example of 3D segmentation after sample stitching.** Visualisation of 3D segmentation of the Plastochron 1 rice leaf primordium from Figure 2 performed using the ITK watershed autoseeded process in MorphoGraphX. Planes were virtually clipped to visualize the segmentation of the inner tissues. Scale bars: 20 µm.

## Data Availability

The .stl file of the holder has been deposited in the Mendeley database at doi: 10.17632/bp6f3bcj5j.1.

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
