# Peer review of "Flip-Flap: A Simple Dual-View Imaging Method for 3D Reconstruction of Thick Plant Samples"

_plants, 2022, doi:10.3390/plants11040506_

Round 1
Reviewer 1 Report
The manuscript describes a method, which the authors dubbed Flip-Flap, that aims to overcome the problem of imaging and 3D reconstruction of thick samples that are often recalcitrant to analyses extending beyond the surface layer. The authors mounted several thick (~300 um) plant samples between two cover slips, placed them in a specially designed and 3D printed coverslip holder and obtained confocal z-stacks from both sides, flipping the sample after one side was imaged. They were then able to stitch the two confocal z-stacks using the open-source software, efficiently generating 3D images of deep tissues and obtaining images of the internal structures suitable for the analysis with programs such as MorphoGraphX. The approach appears to be simple and usable, and should be of interest to many plant biologists.
My one critique concerns the quality of the figures, although this might be due to the journal reducing figure size for review. Fig. 1 in particular is essentially unreadable, and images of the 3D reconstructions probably do not do justice to what can be achieved with this technique. The other figures could probably also be improved (for example, the sizes represented by scale bars are not readable on any of them).
I would also like to see more discussion on how easy it is to align the two stacks: whether there is any shift happening in the x-y dimensions after the sample is flipped that requires more correction than just a simple 180 rotation and whether the z stitching also requires any human input regarding the depth at which the two stacks should be combined.
Reviewer 2 Report
The authors do a good job of showing that this method works on a qualitative level, but given that the flip-flap method is pretty straightforward, I would have hoped that this article would have evaluated the results in a little more detail. In particular, I would have liked to see more about how well the stitching works, a discussion of potential artifacts in the overlap region, or other issues. I don't want to require particular quantitative evaluations, but it would be nice to see more quantitative evaluation as chosen by the authors to give the reader a better sense of how well this works.
Reviewer 3 Report
Specific comments are following:
Figure descriptions need substantial improvement. Each description should be more informative. In particular staining method must be included.
Presented images of the entire structure of the samples do not carry much information, lack of detailes (see fig 1, panel C)
Fig. 1 panel C. The quality of the images is rather poor. No detailes visible.
Fig. 2.Rice root panel. No detailes visible.
Fig. 3 Poor image contrast in the right column.
Methods. Information concerning 3D scanning mode is necessary.
Why the two dies (calcofluor and red direct) with common characteristic had been used? It should be discussed. Which one is better and why? Comparisons should be presented.
Round 2
Reviewer 3 Report
Why 20x and 25x lenses with high numerical apertures were used when the images presented do not contain much details? When small details are not important, it may be worth using lenses with a lower magnification and lower apertures?